# An Update on the Molecular Basis of Phosphoantigen Recognition by Vγ9Vδ2 T Cells

**DOI:** 10.3390/cells9061433

**Published:** 2020-06-09

**Authors:** Thomas Herrmann, Alina Suzann Fichtner, Mohindar Murugesh Karunakaran

**Affiliations:** 1Institute for Virology and Immunobiology, Department of Medicine, University of Wuerzburg, 97070 Wuerzburg, Germany; mohindar.karunakaran@uni-wuerzburg.de; 2Institute of Immunology, Hannover Medical School, 30559 Hannover, Germany; Fichtner.Alina@mh-hannover.de

**Keywords:** γδ T cell, phosphoantigen, BTN, butyrophilin 3, butyrophilin 2A1, evolution, alpaca, human

## Abstract

About 1–5% of human blood T cells are Vγ9Vδ2 T cells. Their hallmark is the expression of T cell antigen receptors (TCR) whose γ-chains contain a rearrangement of Vγ9 with JP (*TRGV9JP* or Vγ2Jγ1.2) and are paired with Vδ2 (*TRDV2*)-containing δ-chains. These TCRs respond to phosphoantigens (PAg) such as (*E*)-4-hydroxy-3-methyl-but-2-enyl pyrophosphate (HMBPP), which is found in many pathogens, and isopentenyl pyrophosphate (IPP), which accumulates in certain tumors or cells treated with aminobisphosphonates such as zoledronate. Until recently, these cells were believed to be restricted to primates, while no such cells are found in rodents. The identification of three genes pivotal for PAg recognition encoding for Vγ9, Vδ2, and butyrophilin (BTN) 3 in various non-primate species identified candidate species possessing PAg-reactive Vγ9Vδ2 T cells. Here, we review the current knowledge of the molecular basis of PAg recognition. This not only includes human Vγ9Vδ2 T cells and the recent discovery of BTN2A1 as Vγ9-binding protein mandatory for the PAg response but also insights gained from the identification of functional PAg-reactive Vγ9Vδ2 T cells and BTN3 in the alpaca and phylogenetic comparisons. Finally, we discuss models of the molecular basis of PAg recognition and implications for the development of transgenic mouse models for PAg-reactive Vγ9Vδ2 T cells.

## 1. Introduction 

Vγ9Vδ2 T cells sense phosphorylated isoprenoid metabolites (Phosphoantigens: PAgs) [1], which are accumulated in the host cell either as a consequence of aberrant or drug-manipulated host cell metabolism or as a result of microbial infection. The subject of this review is recent advances in identifying the molecular mechanism underlying this TCR-mediated sensing and the pivotal role of butyrophilins (BTNs) in this process. A better understanding of these processes can be expected to provide insights into the physiological role of these cells and help to harness them for therapeutic applications as discussed elsewhere. In this review, we outline the current knowledge on the molecular mechanisms underlying phosphoantigen-mediated Vγ9Vδ2 T cell activation, including insights gained by species comparison. In addition, we also aimed to discuss controversial issues and open questions. 

## 2. T Cell Receptors and Antigens

### 2.1. αβ T Cells and γδ T Cells 

Jawed vertebrates (Gnathostomata) [2] possess three lineages of lymphocytes, which are defined by their antigen receptors. These antigen receptors are pivotal for the development of antigen receptor-bearing lymphocytes and the adaptive immune response exerted by them. Antigens are sensed by heterodimeric Ig domain-containing antigen receptors whose diversity is generated by a recombination-activating gene (RAG) 1 and 2-dependent somatic recombination of variable (*V*), diversity (*D*), and joining (*J*) gene segments encoding the antigen-binding variable domain. Different cell lineages can be defined by antigen receptor usage, specifically T cells of the αβ or γδ T cell lineage that express T cell receptors (TCRs) and B cells that produce immunoglobulins (Ig), which serve as B cell receptors (BCRs). Secreted Ig can neutralize antigens, trigger the effector functions of other cells, and activate the humoral components (complement) of the innate immune system. The antigen-binding variable domains (V-domain) of TCRs are encoded by *V*, *D,* and *J* genes of the TCRβ (*TRB)* and TCRδ (*TRD)* loci, and by the *V* and *J* genes of the TCRα (*TRA)* and TCRγ (*TRG*) loci [3]. Each V-domain contains three hypervariable loops (HV) also designated as complementarity determining regions (CDR). The CDR1 and CDR2 are encoded by the *V* genes of the antigen receptor loci. Hence, their diversity is limited by the number of *V* gene segments, while the CDR3 is located at the junction of recombined *V, (D)* and *J* genes and is therefore highly variable. The RAG-dependent recombination mechanism generates CDR3 variability not only by combining *V(D)J* genes but also by junctional diversity ensured by excision or insertion of nucleotides at the recombination sites and insertions of N-nucleotides by the terminal deoxynucleotidyl transferase (TdT). Human TCR δ-chains contain up to three *D* gene-encoded segments, which increases not only the variability of CDR3δ lengths but also massively impacts sequence diversity, since up to four sites of recombination can be incorporated in a CDR3 of a TCR δ-chain [2,4]. For very few species, somatic hypermutations have been described for TCR loci such as the *TRA*, *TRG,* and *TRD* of nurse sharks [5,6] and *TRG* and *TRD* of dromedary [7,8]. Many Gnathostomata, but not mice and humans, possess additional types of RAG-recombined Ig domain-containing antigen receptors that can be considered functional analogues to TCRs or BCRs, respectively [4,9].

### 2.2. Conventional vs. Unconventional T Cells

T cells expressing αβ TCRs, which bind to complexes of polymorphic major histocompatibility (MHC) molecules and peptide antigens (MHC-restricted T cells), are carriers of adaptive cellular immunity. Likewise, T cells with diversified TCR repertoires recognizing antigens in the context of MHC class I-like molecules such as certain types of CD1- or MR1-restricted T cells or even γδ T cells may also exert features of adaptive T cells. The final composition of TCR specificities (repertoire) of MHC-restricted T cells is shaped by intra-thymic positive and negative selection guided by the anatomically controlled presentation of peptide–MHC complexes and the avidity of emerging TCRs to those complexes [10]. A highly conserved but not absolute feature in Gnathostomata is the division of mature T cells into MHC class I-restricted CD8 T cells that exert killer functions and MHC class II-restricted CD4 T cells, which promote and modulate immune functions. Despite a likely co-evolution of the peptide-presenting MHC molecules with *TRA* and *TRB* genes, they cannot be correlated with MHC class restriction or the functional properties of MHC-restricted cells [11]. 

T cells that are not MHC-restricted are commonly described as non-conventional or “unconventional” T cells and can stem from the αβ or γδ T cell lineage. They are also often referred to as “innate T cells” since many of them share features with natural killer (NK) lymphocytes with respect to their susceptibility to antigen-independent signals, especially cytokines, and their expression of NK cell receptors. They differ from conventional T cells in their intra-thymic development and in contrast to MHC-restricted T cells, their TCRs show restrictions in *V* gene usage and unique, characteristic TCR gene rearrangements. Such unique TCR combinations can be used to characterize unconventional T cell populations since they determine, or at least correlate with, a cell type-specific mode of development, functionality, and homing. The best understood populations of non-conventional αβ T cells are CD1d-restricted invariant natural killer T cells (iNKT) cells and MR1-restricted mucosal-associated T cells (MAIT cells). Their α-chains largely carry “invariant” VαJα (*TRAV–TRAJ*) rearrangements and pair with β-chains of limited *TRBV* gene usage. They are specific for certain metabolites bound to the non-polymorphic MHC class I-like molecules CD1d and MR1, respectively [12,13,14]. With regard to γδ T cells, butyrophilins (BTN) [15] or butyrophilin-like molecules, such as SKINT1 [16] in the case of dendritic epidermal cells (DETC), steer the development and activation of certain γδ T cell populations. For some of them, binding in a superantigen-like mode to *TRGV*-encoded parts of the TCR has been demonstrated [17,18,19,20,21].

### 2.3. Vγ9Vδ2 T Cells: TCR and Phosphoantigen Reactivity

The vast majority of human blood γδ T cells are Vγ9Vδ2 T cells (1–5% of blood T cells in healthy individuals), which respond to so-called “phosphoantigens” (PAgs). Their TCRs share a characteristic Vγ9JP (*TRGV9JP*) rearrangement, alternatively designated as Vγ2Jγ1.2 and Vδ2-containing TCR δ-chains. Unless explicitly mentioned, PAg-reactive T cells and Vγ9Vδ2 T cells will be used synonymously in this article. The nomenclature of TCR genes follows that of IMGT [3] and sequence homology to human genes.

Freshly isolated Vγ9Vδ2 T cells share functional features with CD8 T cells and NK cells [22] but under some pathological conditions, TH17-like cells have been observed [23]. Furthermore, at least *in vitro*, they exhibit a remarkable degree of plasticity and multifunctionality such as differentiation into professional antigen-presenting or phagocytosing cells and promoting and regulating immune responses by crosstalk with B cells, dendritic cells, NK cells, and monocytes (reviewed in [24] and schematically shown in Figure 1). Furthermore, the antigen-dependent activation of Vγ9Vδ2 T cells is strongly modulated by additional receptors including inhibitory and activating NK cell receptors [25,26] In the case of NKG2D, even a direct triggering of some effector functions is possible [27].

PAgs are products of isoprenoid synthesis that specifically activate Vγ9Vδ2 T cells. The building blocks of isoprenoid synthesis are isopentenyl pyrophosphate (IPP) and its isomer dimethylallyl pyrophosphate, which are both weak PAgs. The naturally occurring PAg (*E*)-4-hydroxy-3-methyl-but-2-enyl pyrophosphate (HMBPP) stimulates Vγ9Vδ2 T cells about 10,000-fold more efficiently than IPP [28,29,30]. It differs from IPP only in a single hydroxy group and is the immediate precursor of IPP in the non-mevalonate pathway, which is also known as the 2-C-methyl-D-erythritol 4-phosphate/1-deoxy-D-xylulose 5-phosphate or Rohmer pathway. The non-mevalonate pathway is restricted to eubacteria, cyanobacteria, plants, and apicomplexan protozoa. HMBPP is the driving force of a massive Vγ9Vδ2 T cell expansion in infections with HMBPP-producing parasites or bacteria, which can lead to an increase of Vγ9Vδ2 T cells from 1–5% of blood T cells to more than 50%. With the exception of apicomplexan parasites such as *Plasmodium spp.* or *Toxoplasma gondii*, all animals synthesize IPP exclusively via the mevalonate pathway [29]. Reduction of the activity of the IPP-metabolizing enzyme farnesyl-diphosphate-synthase (FPPS), e.g., by inhibitors such as aminobisphosphonates (ABP) and zoledronate (Zol) [31,32] or shRNA-mediated knock-down of FPPS expression [33], reduces the metabolization of IPP and leads to a concomitant increase of IPP levels. Human or primate antigen-presenting cells (APC) or target cells with increased IPP levels are sensed by the Vγ9Vδ2 TCR and Vγ9Vδ2 T cells are activated. The activation by triphosphate nucleotide derivatives, naturally occurring as a consequence of increased IPP and HMBPP levels, has also been reported [34].

Some tumors such as the human B cell lymphoma Daudi spontaneously activate Vγ9Vδ2 T cells [35]. This activation depends on the intracellular accumulation of IPP and can be abolished by statins, which inhibit the 3-hydroxy-3-methylglutaryl-CoA reductase (HMG–CoA) reductase and consequently also IPP synthesis [32]. However, it is unclear how PAg action relates to the reported binding of the Vγ9Vδ2 TCR to some cellular proteins, as in the case of TCR G115 binding to ectopically expressed F1-ATPase [36] or the binding of other Vγ9Vδ2 TCRs to stress-associated molecules such as ULBP4 [37], MutS homologue 2 [38] or hsp60/GroEl [39].

In summary, Vγ9Vδ2 T cells can sense the metabolic changes of transformed [32], infected [40], [29] or drug-treated host cells via their TCR [31,32]. This reactivity can be harnessed clinically as the remission of certain tumor entities after Vγ9Vδ2 T cell activation has been observed (reviewed in [41,42]). Furthermore, Vγ9Vδ2 T cells can support anti-bacterial immunity in preclinical mouse and primate models, including the possibility of PAg-based vaccines against tuberculosis as shown for non-human primates [43,44]. 

## 3. Butyrophilin 3 (BTN3) as PAg Sensor: Identification and Functional Analysis

### 3.1. Murine Reporter Cells Identify the Crucial Role of Human Chromosome 6 (Chr:6) in Vγ9Vδ2 T Cell Activation by PAg

PAgs activate Vγ9Vδ2 T cells very rapidly as documented by the acidification of culture medium already nine seconds after the addition of the synthetic PAg BrHPP to a cell culture with a Vγ9Vδ2 T cell clone [45]. This finding has originally been taken as a hint for a direct binding of the PAg to the TCR, but attempts to demonstrate such an interaction have not been successful, despite the identification of positively charged hypothetical binding sites for PAg in the G115 Vγ9Vδ2 TCR crystal that resemble binding sites originally identified in phospholipids antibodies [46]. The idea of PAg directly binding to the TCR was in conflict with the increasing experimental evidence for a presentation of PAgs to the Vγ9Vδ2 T cell by cells of human or primate origin [47], leading to a concept of non-polymorphic species-specific molecules acting as direct PAg-presenting molecules or being otherwise mandatory for PAg stimulation, e.g., by providing a γδ T cell-specific co-stimulus [48,49,50]. Essentially, all cell types, including human γδ T cells and the widely used Jurkat T cell lymphoma, have the capacity of PAg presentation [47]. Furthermore, the activation of primary γδ T cells is massively modulated by activating and inhibiting cell surface receptors, leading to an interference that should be avoided [25,26,27]. We hypothesized that rodents would lack PAg-presenting molecules and that rodent reporter cell lines would not be reactive to xenogeneic NK cell receptor ligands. Therefore, we generated reporter cells of murine origin (mouse 58C or mouse-rat T cell hybridoma 53/4) transduced with a Vγ9Vδ2 TCR of proven PAg reactivity to assess the presence of a putative PAg-presenting or Vγ9Vδ2 T cell-(co)stimulating molecule for cells of various tissue and species origin (Figure 2). Surprisingly, although interleukin (IL)-2 production by the TCR transductants could be triggered by anti-CD3 or anti-TCR monoclonal antibodies (mAbs), initial attempts to generate an IL-2 response to PAg in co-cultures with cells of human or primate origin failed [51]. However, this lack of response could be rescued by providing a co-stimulatory signal through the overexpression of a rat/mouse CD28 construct in the reporter cell and the use of CD80/CD86-expressing stimulatory cells; in some cases, this was achieved by CD80 transduction [51,52]. The generation of these reporter cell lines allowed to interrogate multiple cell types for their capacity to present PAgs. To reduce the chance of false-negative results, the same type of reporter cells was also tested for an αβ TCR response to a peptide antigen presented by APCs transduced with rat MHC class II molecules [53,54]. All types of cells of human or primate origin induced a PAg-dependent IL-2 production, but no response was observed with cells from mice, rats, hamster, dog, and cow (Kreiss, Li, Herrmann, Karunakaran, unpublished data). 

### 3.2. The Human Butyrophilin 3 (BTN3A) Family

Game-changing for understanding the molecular basis of Vγ9Vδ2 T cell activation by PAgs was the identification of human BTN3A molecules as key compounds in PAg-induced Vγ9Vδ2 T cell activation [53]. In humans, the *BTN3A* gene family consists of *BTN3A1*, *BTN3A2,* and *BTN3A3,* which are part of a *BTN* gene cluster at the telomeric end of the MHC complex on Chr:6 [55]. Antibodies raised against the BTN3A1 (CD277) extracellular domain (ED) are available but cross-react with other members of the BTN3A family [56,57]. BTNs are named after BTN1A, which is a membrane protein that is involved in fat droplet formation in milk and displays an immunomodulatory potential similar to many other BTNs and BTN-like molecules (BTNL) [58]. BTN3A1, similar to most BTNs, carries an extracellular domain with strong structural similarity to B7 receptor family molecules consisting of an N-terminal IgV-like domain (V domain) followed by an IgC-like domain (C domain), a transmembrane domain (TM), and an intracellular domain (ID) [59]. The ID contains a juxtamembrane domain (JM) followed by a B30.2 or PRY/SPRY domain and a tail consisting of variable numbers of amino acids (aa) with unknown function (Figure 3). B30.2 [55,58] is found in many molecules involved in innate immunity such as tripartite motif (TRIM)-containing proteins [60]. Some BTN family members such as myelin-oligodendrocyte glycoprotein (MOG), the BG molecules of chicken or the selection and upkeep of intraepithelilial T-lymphocyte protein (SKINT) and SKINT-like molecules share B7-like extracellular domains (or parts of it) but vary considerably in their transmembrane/intracellular parts [59].

The EDs of BTN3A1, A2 and A3, are highly homologous. Their V domains differ by a single, conservative substitution (R37K), and their C domains are also quite similar, while the TMs and IDs of BTN3As differ considerably [59]. Interestingly, the B30.2 of BTN3A1 and BTN3A3 are more similar to each other than their TM and JM. The ID of BTN3A2 lacks the B30.2 and a part of the JM [59] (aligned in Figure 3). BTN3-specific antibodies act as co-stimulators for γδ T cells, but BTN3 has also been implicated in the negative and positive regulation of NK cell and monocyte responses [56,61,62,63].

### 3.3. BTN3A1 is Mandatory for Vγ9Vδ2 T Cell Activation 

The key finding in the identification of BTN3A1 as a mandatory component for PAg-mediated Vγ9Vδ2 T cell activation was the induction of cell proliferation by the agonistic BTN3A-specific mAb 20.1 and the inhibition of PAg-mediated activation by mAb 103.2. Both mAbs bind to different sites of the V domain of BTN3A molecules: mAb 20.1 to the C, C´, and C strands and the B-C, C´-C´´, and D-E connecting loops (Figure 3) [57]. In their landmark paper, Harly et al. described not only the general importance of BTN3A molecules for the PAg response but also that BTN3A family members differentially support PAg-dependent Vγ9Vδ2 T cell activation [53]. 

The nearly ubiquitous expression of BTN3A, which includes γδ T cells, the stimulatory properties of mAb 20.1 for γδ T cells, and the B7-like nature of BTN3A-EDs, raised the question of whether BTN3 molecules expressed by γδ T cells themselves or those expressed by the “presenting” cells mediate activation. This was addressed by pulsing antigen-presenting or target cells with mAb 20.1 or with the ABP pamidronate and testing mAb 20.1- or PAg-mediated activation with our murine Vγ9Vδ2 TCR-expressing reporter cells, which lack BTN3 molecules and do not “present” PAg [53]. Since mAb 20.1 and 103.2 bind to all three BTN3A molecules, it was important to test the contribution of the different BTN3A molecules to PAg reactivity. To this end, primary Vγ9Vδ2 T cell lines [56,57] were stimulated with pamidronate-pulsed BTN3A isoform-specific knock-down cells and *BTN3A* knock-down cell lines transfected with single *BTN3A* genes. These experiments [53] revealed an essential role of BTN3A1 in APB-induced stimulation but not of BTN3A2 and BTN3A3 [53,67,69,70], whose important contribution to the magnitude of PAg responses was demonstrated later by others [20,71,72].

The structural characterization of soluble and crystalized recombinant BTN3A-EDs showed only small differences among the three BTN3As but identified two types of BTN3A1 homodimers: one dimer with a symmetric parallel V-shaped structure with interacting C domains and another asymmetric head-to-tail conformation with contact of the V and C domain (Figure 4) [57]. The agonistic mAb prevents the formation of the head-to-tail conformation, arguing for the V-shaped conformation as the biologically active form and the head-to-tail conformation as the inactive one [57]. Another finding was an ABP-induced immobilization of BTN3A1, but not of the non-stimulatory BTN3A2 or BTN3A3, which led to the notion that this type of immobilization might be crucial for BTN3A-mediated stimulation [53]. However, this view was revised after the demonstration that the introduction of a disulfide bridge between the C domains of two BTN3A1s, which prevents the head-to-tail conformation, diminished ABP- and mAb 20.1-induced stimulation but not BTN3A1 surface immobilization [73]. In accordance with this, Yang et al. proposed the head-to-tail conformation of BTN3A1-ED as the biologically active one [74]. Nevertheless, the conformation of the activating state of BTN3A1 is still a matter of debate as illustrated by an editorial on the same study depicting a model with an “activated” BTN3A1 in a V-shaped conformation [75]. To our knowledge, the possibility that a reversible switch between the two conformations or co-existence of both is required for achieving the activating state of BTN3A or BTN3A-associated complexes has not been discussed yet. 

### 3.4. BTN3A1 Acts as a PAg Sensor

The importance of the intracellular domain of BTN3A1 for PAg sensing was already demonstrated by Harly et al. who showed that in contrast to BTN3A3, a BTN3A3 containing the BTN3A1-ID induced ABP-mediated Vγ9Vδ2 T cell stimulation [53]. PAg binding to the B30.2 domain has been determined by a number of methods: Isothermal titration calorimetry (ITC) [20,64,67,71,76], nuclear magnetic resonance (NMR) spectroscopy [64,76,77], small-angle X-ray scattering (SAXS) [64], and fluorescence polarization [78]. Affinities for HMBPP were in the micromolar range and in the millimolar range for IPP. 

Crystallography originally identified a shallow binding groove in the apo-form [71] or the formaldehyde-fixed BTN3A1–B30.2 domain-PAg complex [67], which could accomplish PAg binding and later in complexes of B30.2 domains with various PAgs [74]. NMR studies revealed chemical shift perturbations (CSP) within and close to the PAg binding site [76], but also in more distant areas of B30.2 and in the JM domain [64,76,77]. As in the case of the ED, two types of B30.2 dimers have been observed: a symmetric one in which the PAg binding sites are positioned distant from each other, which was designated as dimer II by Gu et al. [73] or pattern A by Yang et al. [74], and another one with one binding site pointing to the interface of both domains and the other binding site to the outside (dimer I [73] or pattern B [74], respectively) (Figure 5). A key residue is histidine H351, which changes its conformation as a consequence of PAg binding [74]. Its importance is demonstrated by the fact that after R351H substitution, BTN3A3 becomes an efficient PAg sensor [67]. Interestingly, the 1-OH group of the exogenous PAg HMBPP, which is missing in the much weaker IPP, forms additional H-bonds with H351 and tyrosine Y352 and might explain the differential binding and biological activity of both PAgs [73,74]. Furthermore, HMBPP binding favors the interaction of H351, W391, and W350 with the juxtamembrane region in pattern B [73,74] and mutagenesis of W350, which is part of the dimer I/pattern B interface that completely abrogated the PAg response of Vγ9Vδ2 T cells but did not affect PAg binding to B30.2 [74]. A final argument in support of the importance of dimer B for T cell activation stems from the analysis of HMBPP derivatives where the methyl group of HMBPP was replaced by other residues, including a bulky 4-methylbenzyl (HMBPP-08). This molecule binds more efficiently to the B30.2 domain than HMBPP but is much less active (EC_50_ 198 nM versus 0.016 nM). This lower biological activity is unlikely to reflect impaired transport across the plasma membrane since the hydrophobic nature of HMBPP-08 would favor passing the membrane as previously shown for other highly active HMBPP derivatives [74]. Furthermore, the permeabilization of cells with monensin did not change activation. Therefore, the authors argue that the loss of activation probably results from the bulky nature of the 4-methylbenzyl, which inhibits the formation of a regular pattern B dimer and leads to a new B30.2 conformation identified in HMBPP-08-B30.2 co-crystals [74].

It is unclear whether and how changes in B30.2 and JM translate to the ED and to the interaction between BTN3 and TCR. Yang et al. [74] addressed this by atomic force microscopy and measured mechanical forces of the cell–cell interaction of ABP-pulsed or unpulsed pancreatic tumor and Vγ9Vδ2 T cells. This force increased five minutes after contact of the ABP-pulsed presenting cell and was interpreted as evidence for BTN3A1–TCR binding [74]. However, this interpretation should be met with caution, as increased mechanical force could also reflect a general increase of cell–cell adhesion, e.g., as a consequence of TCR-mediated inside-out integrin activation [75].

### 3.5. The Importance of the Juxtamembrane Domain

The BTN3A1–JM is not only important for PAg sensing but also crucial for the formation of complexes with other molecules such as periplakin and the small GTPase RhoB [71,79]. Periplakin is a large protein (1756 aa) whose cytosolic domain binds to membrane proteins with its N-terminal plakin domain and links them with its C-terminal domain to intermediate filaments. It was co-immunoprecipitated from lysates of BTN3A1 and of BTN1A1 transductants whose BTNs share a 7 aa peptide motif encoded by exon 5 [71]. Deletion of this domain or mutation of a local dileucine abolished ABP-mediated stimulation [71]. Nevertheless, the role of periplakin in the activation is unclear, since knock-down experiments for periplakin led to no conclusive results [71,80]. Furthermore, the periplakin binding motif is missing in alpaca BTN3, which efficiently mediates PAg stimulation [20]. The other BTN3A1-binding molecule proposed to participate in PAg stimulation and to bind to the JM is the small GTPase RhoB. RhoB became a candidate molecule for controlling ABP responses after analysis of Interferon (IFN)-γ production in Vγ9Vδ2 T cells in cultures with pamidronate and Epstein Barr Virus (EBV)-transformed B cell lines obtained from several family pedigrees [79], where 33 cell lines induced activation and seven did not. One of the three candidate genes identified by its vicinity to single nucleotide polymorphic markers was the small GTPase RhoB. ABPs are known to increase the levels of the GTP-bound form of Rho-GTPases. The modulators of Rho activity modulated Vγ9Vδ2 T cell stimulation and led to changes in the ABP-induced reduction of BTN3A1 mobility [79]. The specificity for RhoB was shown by experiments with constitutively active or dominant-negative RhoB mutants and CRISPR/Cas9-mediated knock-out cells for RhoA, -B, and -C. Only RhoB deficiency reduced APB-mediated IFN-γ stimulation, suggesting a connection between RhoB and Vγ9Vδ2 T cell stimulation [79]. A mechanistic explanation for this could be that activated RhoB links BTN3A1 to the cytoskeleton, which would fit to the ABP treatment-correlated reduction of surface mobility observed for BTN3A1. Furthermore, RhoB binds to recombinant BTN3A1-ID but not to B30.2 alone, suggesting the involvement of the JM in this process. The authors show by biolayer interferometry that in a second step and as a consequence of PAg binding, BTN3A1 is released from RhoB. Accompanied with ABP treatment is also a new BTN3A1–ED conformation demonstrated by Förster resonance transfer (FRET) with antibodies against the BTN3A1 V domain and 4,4-difluoro-4-bora-3a,4a-diaza-s-indacene (BODIPY)-labeled cells, which showed an increased distance between V domain and plasma membrane as expected for the adaptation of a V-shaped conformation of the ED [79]. 

Although the importance of the JM for BTN3A1 action is undisputed, its conformation and conformational changes upon PAg binding are not understood. The Morita group performed extensive molecular modeling-based analysis, complemented with mutational analysis and a comparison of BTN3 sequences of species possessing (putatively) PAg-reactive Vγ9Vδ2 T cells and favor a coiled-coil structure of the JM of the BTN3 dimer [65]. In this model, the JM would constantly maintain a certain distance between the B30.2 domain and the cell membrane which would demand a head-to-tail dimer of the ED suggested by Yang et al. [74]. Alternatively, small-angle X-ray scattering of recombinant BTN3A1–IDs suggests a more globular structure, which becomes more compact upon PAg binding. As in the case of the ED dimers, there is still no consensus regarding which “active” formation is adopted after PAg binding. As discussed for the ED, it might well be that phosphoantigen sensing requires a reversibility or co-existence of conformations. In this case, all means (e.g., mutations) of fixing one conformation would negatively impact PAg-mediated activation. Another important aspect is the role of the JM as a possible interface for BTN molecules or other, not yet defined, binding partners which may also be affected by or modify the JM conformation. 

### 3.6. Cooperation of BTN3 Isoforms 

While the mandatory role of BTN3A1 for PAg sensing was readily confirmed in all knock-down studies, the knock-down experiments of BTN3A2 and BTN3A3 led to variable results. The first clear negative effects were reported on ABP and HMBPP responses by Rhodes et al. who found a complete loss of stimulation by BTN3A1 knock-down but also clear effects of BTN3A2 and BTN3A3 knock-down [71]. These were confirmed and extended by CRISPR-Cas9 knock-out 293T cell lines by Vantourout and colleagues [72]. Various combinations of BTN3A knock-out and re-expression were tested for their capacity of Zol-induced stimulation of Vγ9Vδ2 T cell lines. Knock-out of BTN3A2 and BTN3A3 abolished the ABP-mediated stimulation of Vγ9Vδ2 T cell lines. Knock-out of BTN3A2 alone reduced it significantly, while the effects of BTN3A3 knock-out were less pronounced [72], which is in line with our unpublished data obtained by testing HMBPP responses of γδ TCR-MOP-transduced murine reporter cells, except for BTN3A3 knock-out, which also exhibited a remarkable reduction in the PAg activation of reporter cells. Another notable difference concerning BTN3A3 was that Vantourout et al. found a residual response to cells expressing only BTN3A3 or BTN3A knock-out lines reconstituted with BTN3A3. However, in our hands, the knock-out of all BTN3A isoforms (BTN3AKO) and transduction with BTN3A3 did not restore any HMBPP-mediated activation of murine TCR–MOP transductants (Karunakaran et al. unpublished data). Whether this reflects a peculiarity of the different assay systems or responses to Zol versus HMBPP remains to be tested.

Altogether, these studies confirmed the pivotal function of BTN3A1 in PAg sensing but also demonstrated a clear positive effect of the other two BTN3A proteins on BTN3A1 action, with BTN3A2 being more efficient in this respect. BTN3A1 and BTN3A2 probably interact directly, as concluded from a 1:1 stoichiometry in co-immunoprecipitation experiments, common intracellular localization, and trafficking. Both molecules were retained in the ER, although at a different degree, but the formation of BTN3A1–BTN3A2 heterodimers released them from this retention and led to an increased cell surface expression while requiring the C domain of BTN3A2 [72]. Nevertheless, this overexpression does not explain the increased Zol- or HMBPP-induced stimulation, since mutations leading to increased cell surface expression do not necessarily result in increased activation [72]. Mutational analysis of an ER retention motif within the JM and of putative interaction sites between both molecules within the JM allow the conclusion that trafficking, as well as the control of ER retention is instrumental for efficient Zol-induced stimulation. An important technical aspect when interpreting such experiments are diverging effects of tagging BTN3A proteins and the generation of BTN3A-fusion constructs on cellular trafficking and function [72]. Examples are the fusion with C-terminal EGFP which, although leading to increased cell surface expression of BTN3A1, reduced stimulation effects while small N-terminal tags (FLAG and hemagglutinin) had no effect [72]. In addition, no negative effects on stimulation were seen for the fusion proteins of human BTN3A1 or alpaca BTN3 and C-terminal mCherry protein [20]. 

The preferential formation of BTN3 heterodimers over BTN3A1 homodimers fits well with the findings of Gu et al. who expressed and co-purified full-length recombinant BTN3A1 and BTN3A2 molecules and isolated only BTN3A1–BTN3A2 heterodimers [73]. In line with this was our unpublished data that also favors BTN3A heterodimers rather than BTN3A1 homodimers (Karunakaran et al. unpublished data). 

In summary, the data discussed so far highlight the JM as a candidate region for explaining the higher efficiency of PAg sensing by BTN3 heteromers over BTN3A1 alone, which affects intracellular trafficking and surface expression and likely contributes to yet unknown events such as the recruiting of additional molecules. Thus, the JM region of the BTN3A1 molecule is crucial for PAg sensing but in addition, the JM of BTN3A2 and BTN3A3 in BTN3 heteromers enhance this capacity. Interestingly, at least one function of BTN3A1 is exclusive to BTN3A1 but without obvious links to Vγ9Vδ2 T cell stimulation [81]. BTN3A1, but not BTN3A2 or BTN3A3, controls the induction of type I interferon responses by cytosolic nucleic acids and viruses via its ID [81], whereby the intracellular domain forms a complex with TANK-binding kinase (TBK1) and microtubule-associated protein 4 (MAP4). After stimulation with nucleic acids, dynein displaces MAP4 and redirects the BTN3A1–TBK1 complex within the cell to a perinucleic region. Finally, BTN3A1 mediates the interaction of TBK1 with interferon-responsive factor 3 (IRF3), leading to the phosphorylation of IRF3, its translocation in the nucleus, and the positive regulation of type I interferon production. 

### 3.7. The Role of TCR Clonotypes in the Response to mAb 20.1 and PAg

Stimulation of human peripheral blood mononuclear cells (PBMCs) with mAb 20.1 or its sc-Fv induces a robust activation of Vγ9Vδ2 T cells or Jurkat cells transduced with the γδ TCR-G115 [57]. Furthermore, mAb 20.1 and derivatives thereof can trigger anti-tumor responses in preclinical models of Vγ9Vδ2 T cell-mediated tumor therapy [82,83]. We also observed the activation of TCR–MOP transductants by mAb 20.1 in the presence of the human B cell lymphoma RAJI, although not to the same degree of PAg stimulation [53]. This seems to be in conflict with the observation that Vγ9Vδ2 γδ TCR–D1C55 transgenic mouse cells were stimulated in cultures with human cells and HMBPP but not with mAb 20.1, which even inhibited the PAg response. For a better understanding of these results, we compared the activation of the different TCRs using the same reporter system (the murine reporter cells introduced above) and human RAJI cells as APCs and found that all Vγ9Vδ2 TCR-transduced reporter cells responded very similar to HMBPP [54]. However, in contrast to TCR–MOP, the response of TCR–G115 and TCR–D1C55 transductants to mAb 20.1 was very weak or not detectable, and similar results were obtained with sc-Fv fragments of both mAb. Other key findings were that TCRs containing γ- and δ-chains of TCR–MOP and TCR–D1C55 induced some reduction in the PAg response and a strong reduction in the mAb 20.1 response of reporter cells. The residual response could be largely attributed to the TCR–MOP γ-chain. Titration experiments with the different TCR transductants and different concentration of PAg and mAb 20.1 as well as 20.1 sc-Fv showed that both mAb 20.1 and 20.1 sc-Fv inhibited the PAg response. In the presence of saturating concentrations of mAb 20.1, PAg-induced stimulation was never below that induced by the mAb alone. To explain in brief, stimulation with a saturating concentration of mAb 20.1 alone led to the production of 100 ng/mL IL-2 by one specific TCR transductant cell line. In this case, mAb 20.1 reduced the stimulation by a high PAg concentration from 1000 to 100 ng/mL (90% inhibition) and in the presence of a low PAg concentration from 200 to 100 ng/mL (50% inhibition). This explains why TCR-D1C55, which hardly induced a mAb 20.1 response, resulted in a nearly complete inhibition of the PAg response [54,66], while the 20.1 mAb-responsive TCR-MOP transduced cell line showed a significant residual PAg-independent response [54]. Finally, we also tested the mAb response of transduced Jurkat cells and could reproduce the clonotypic differences for 20.1 but not for HMBPP- or Zol-induced CD69 activation. However, differences were much less pronounced, which may reflect a lower activation threshold either of this reporter cell type or of the assay [54]. In the future, it will be interesting to study how much the clonotypic differences affect the response of primary cells. A mechanistic explanation for the differential activity of mAb 20.1 is still missing, but we suggest two mutually non-exclusive possibilities. The first one would be a function of mAb 20.1 as a partial agonist, which would “freeze” the activating stage at a certain point and by this prevent complete activation, while the other one would be that the 20.1 binding surface formed by the BTN3A V domain partially overlaps with the site relevant to BTN3A action [54] for PAg reactivity, as it will be discussed later. 

## 4. Lessons from Armadillo and Alpaca

### 4.1. Vγ9Vδ2 TCRs and BTN3s in Different Species

Species comparison can open new perspectives for understanding the minimal molecular requirements of body functions, e.g., the detection and elimination of pathogens, by comparing molecules among different species. 

In the case of PAg-specific T cells, we aimed to identify those cells in non-primate species hoping to find minimal molecular signatures of the PAg sensing system [84,85,86]. To this end, genomes of species originally sequenced as part of the 24-mammalian genome project and other accessible genomes were tested for genes encoding the triad of Vγ9 (*TRGV9*), Vδ2 (*TRDV2*), and the ED of BTN3. Apart from Old and New world monkeys, which were already known to respond to PAg similar to humans, only a small number of species from very different phylogenetic groups carrying all three genes were tested for open reading frames (ORFs). In some of these species such as cows and horses, those genes contained single stop codons or substitutions, which are likely to result in the loss of function, e.g., by disruption of the Ig domain disulfide bridge. In other species, no homologues of *TRGV9*, *TRDV2,* and *BTN3–ED* genes were found [84,85,86]. This is consistent with lacking reports on PAg-reactive cells in rodents and our own failed attempts to find such responses in mice and rats [84,85]. The *BTN3–ID* genes were not assessed at this stage due to a high number of B30.2-containing genes and possible homologues The non-primate species that were finally found to contain functional ORFs of the PAg-sensing triad of *TRGV9, TRDV2,* and *BTN3* genes were camelids and whales. Important for the phylogeny of Vγ9Vδ2 T cells was the conservation of *TRGV9-, TRDV2-,* and *BTN3*-like genes in armadillo and sloth. Both belong to the order of Xenarthra, which is part of the Eutheria magnaorder Atlantogenata, while other species possessing *TRGV9-, TRDV2-,* and *BTN3*-like genes belong to the Boreoeutheria, which is the other magnaorder of Eutheria. This implies that the common ancestor of Eutheria and of all placental mammals possessed those genes. An argument for a functional relationship and probable co-evolution is that nearly all species with an ORF for *BTN3* also possess ORFs of *TRGV9-* and *TRVD2*-like genes [84,85,86].

### 4.2. Armadillo: A Witness of Vγ9Vδ2 T Cell Evolution

The analysis of the armadillo genome revealed ORFs for *TRGV9*, *TRDV2,* and *BTN3-ED,* which was of interest not only for phylogenetic reasons. First of all, armadillos serve as a natural reservoir and model organism for infections with *Mycobacterium leprae* [87], and secondly, human Vγ9Vδ2 T cells have been implicated in the defense to mycobacterial diseases [88]. In addition, it was reported that *M. leprae*-infected animals showed an increased frequency of circulating lymphocytes stained by an antibody specific for human TCR δ-chains [89]. To our disappointment, the analysis of cDNA from armadillo PBMCs revealed neither transcripts for *BTN3* nor *TRGV9,* although fragments of the genes could be amplified from genomic DNA and analysis of genomic sequences gave no apparent reasons for loss of functionality [90]. A closer inspection of the armadillo genome identified three *BTN3* genes. For none of them exons for a leader peptide were found, while all possessed exons for the V and C domain, although not fully translatable. Exon 4, which encodes the connecting peptide between the C domain and TM in human BTN3A and vpBTN3, was found in two *BTN3*-like genes. Exons encoding sequences of the JM were mostly lost, but the respective introns were identified by homology to human introns. The most complete *BTN3* gene contained an exon for B30.2 but with an internal stop codon. Nevertheless, codons for amino acids that form the PAg binding site of huBTN3A1 and vpBTN3 B30.2 domains were conserved. Consequently, the common ancestor of placental mammals probably possessed BTN3(s) with PAg-binding properties [90]. 

Interestingly, TCR δ-chains with *TRDV2J4* gene rearrangements could be cloned, expressed, and paired with a human *TRGV9JP*-containing TCR γ-chain. This was demonstrated by the cell surface staining of mouse CD3ε and *TRGV9* after co-transduction of a mouse T cell hybridoma with genes for armadillo *TRDV2*-containing TCR δ-chains and a human Vγ9 TCR chain. These cells produced IL-2 after stimulation with immobilized anti-mouse CD3 and anti-Vγ9 mAb but not in co-cultures with HMBPP and human APCs [90].

In contrast, the dolphin genome possesses two *BTN3-*like genes. One non-functional gene covering the BTN3–ED exon and one full-length BTN3 ORF, which showed the highest aa identity with human BTN3A3 and vpBTN3 and also a full conservation of the amino acids of the PAg binding site [90]. This, together with the detection of in-frame *TRGV9JP* rearrangements and *TRDV2* genes, renders the bottlenose dolphin a prime candidate for PAg-reactive Vγ9Vδ2 T cells [91]. 

### 4.3. Alpaca: The First Non-Primate Species with PAg-Reactive Cells

Among the non-primate candidate species for PAg-reactive Vγ9Vδ2 T cells, alpaca was the only species accessible to us. Analysis of PBMCs confirmed the genomic sequences in the database and the expression of the ORFs. Comparable to the armadillo, nearly all *TRDV2* sequences were rearranged with *TRDJ4* gene segments whereas, in humans, *TRDV2* rearrangements were dominated by *TRDJ1* gene segments, and *TRDJ4* gene segments are rare [92,93]. Nearly all *TRGV9* were rearranged with *TRGJP*, and the frequency of productive in-frame rearrangements was similar to humans. Three, probably allelic, *TRGJP* sequences (*JPA, JPB, JPC)* were identified. Interestingly, *vpBTN3* occurs as a singleton and had higher sequence similarity to *BTN3A3* compared to *BTN3A1* but complete identity in the amino acids defining the PAg binding pocket of BTN3A1 [85,86]. ITC studies of recombinant wild-type and mutant BTN3 B30.2 domains showed very similar binding characteristics for HMBPP and IPP to huBTN3A1 and vpBTN3 [20]. 

To test for a BTN3-dependent PAg response of alpaca Vγ9Vδ2 T cells [20], we generated monoclonal antibodies against alpaca Vγ9Vδ2 TCRs and vpBTN3 using transduced mouse cells as immunogens. Two monoclonal antibodies were characterized in greater detail: the TCR-specific mAb WTH-4 and the BTN3-specific mAb WTH-5. WTH-4 bound to transductants expressing alpaca Vγ9Vδ2 TCRs and to TCRs composed of human Vγ9 and alpaca Vδ2 TCR chains but not to human Vγ9Vδ2 TCRs, suggesting the localization of the epitope on the alpaca Vδ2 chain. Furthermore, WTH-4 was specific for transductants expressing alpaca Vγ9Vδ2 TCRs with a *TRDV2J4* but not *TRDV2J2* rearrangements. Frequencies of WTH-4-positive cells ranged from 0.2% to 1.2% of CD3-positive lymphocytes among individual animals but also sometimes varied between time points of blood sampling [20]. 

The mAb WTH-5 stained vpBTN3-transduced murine and human cell lines. Given the high similarity of alpaca and human BTN3-EDs, both BTN3s were compared for binding of the mAbs originally raised against vpBTN3 (WTH-5) and BTN3A1 (20.1 and 103.2) by staining transduced CHO cells and primary blood cells. WTH-5 stained vpBTN3-transduced CHO cells very efficiently and showed some degree of cross-reactivity to CHO or 293T cells transduced with human BTN3A1 at high antibody concentrations. mAb 103.2 showed a converse staining pattern: a good staining of human BTN3A1 transductants and poor staining of vpBTN3-expressing cell lines. When tested on PBMCs, mAb 103.2 stained only human cells and mAb WTH-5 stained only alpaca cells. Phycoerythrin (PE)-labeled mAb 20.1 was weakly cross-reactive for vpBTN3-transduced CHO cells but when tested on PBMCs, binding to alpaca lymphocytes and monocytes reached more than half of the intensity (geometric mean fluorescence intensity) of the respective human cells. Surprising was also the staining pattern of mAb 103.2 on human PBMCs. MAb stained human lymphocytes well, but the epitope was barely detectable on monocytes although PE-labeled mAb 20.1 bound to both cell types similarly. Given that mAb 103.2 efficiently antagonizes PAg stimulation, this observation warrants further analysis and might indicate a cell type-specific conformation or modification of the molecule hiding the 103.2 epitope [20].

The new monoclonal antibodies allowed us to directly trace HMBPP reactivity in primary cell cultures. WTH-4-positive cells expanded in a dose-dependent fashion in cultures with HMBPP and IL-2, and this expansion was blocked by the BTN3-specific mAb WTH-5. Single-cell PCR of HMBPP cultures showed that essentially all WTH-4-positive cells expressed *TRDV2J4* rearrangements. Analysis of cloned RT-PCR products and sequencing of clones with *TRV*- or *TRC*-specific primers revealed no specific sequence motifs for CDR3s of PAg-responsive cells but a homogenization of *TRGV9JP* CDR3 lengths, similar to human PAg-reactive Vγ9Vδ2 T cells. Notably, a dominance of single clones with a *TRDVJ2* rearrangement among HMBPP-stimulated WTH-4-negative cells was observed. 

To directly prove the PAg reactivity of Vγ9Vδ2 TCRs, TCR sequences were cloned and transduced in murine reporter cells (described above). TCRs cloned from single WTH-4-positive cells had *TRGV9JP* and *TRDV2J4* rearrangements, while WTH-4-negative HMBPP-expanded cells possessed *TRGV9JP* and *TRDV2J2* sequences. These TCR transductants were compared for their response to PAg and mAb 20.1 with reporter cells expressing human TCRs (TCR–MOP) and TCRs containing γ− and δ-chains cloned from total alpaca PBMCs. TCR–MOP-expressing cells responded well to HMBPP and mAb 20.1 in the presence of 293T cells but not in the presence of 293T cells where all three *BTN3A* genes were inactivated by CRISPR–Cas9-induced mutations (BTN3KO 293T cells) transduced with vpBTN3. The alpaca TCRs (vpTCRs) cloned from HMBPP-expanded WTH-4-positive and -negative single cells induced a reporter cell response to HMBPP presented by 293T or vpBTN3-expressing BTN3KO 293T cells. Previously studied TCRs, containing randomly cloned sequences of Vγ9 and Vδ2 chains of unstimulated PBMCs [84] resulted in no PAg response at all, emphasizing the importance of the right combinations of Vγ9 and Vδ2 TCR chains for an efficient PAg response. Despite the reactivity of the alpaca Vγ9Vδ2 TCR-transduced cells to PAg presented by wild-type 293T cells, stimulation by mAb 20.1 was not observed, and this was also not the case in primary alpaca cell cultures. Since culture conditions of primary cells were not optimal (alpaca cells died after 6–7 days of culture), and since mAb 20.1 responses largely depended on the CDR3s of the Vγ9Vδ2 TCR [94], it is so far not possible to state whether this unresponsiveness is a feature of alpaca Vγ9Vδ2 TCRs in general or of the tested clonotypes [20]. 

The differential specificity of alpaca Vγ9Vδ2 TCR transductants that recognize PAg in the context of alpaca as well as human BTN3 and human TCR–MOP transductant, which responds exclusively to cells expressing human BTN3, will give an opportunity to identify BTN3 and TCR regions controlling PAg-mediated activation by comparing inter-species chimeras or single amino acids mutants and identify aa sequences that are crucial for activation [20].

### 4.4. Alpaca BTN3: An All-in-One Solution

In contrast to humans, alpaca possesses a *BTN* cluster with single copies of *BTN1*, *BTN2,* and *BTN3*. The singleton nature of vpBTN3 implies that the capacity of PAg sensing and γδ T cell activation is merged in vpBTN3 while humans require the cooperation of two to three/several BTN3As. To identify which parts of the BTN3 molecules “help” during PAg sensing, chimeras of human BTN3A1 and alpaca BTN3 were expressed in BTN3KO (293T cells with CRISPR-Cas9 inactivated *BTN3A1, A2* and *A3* genes) or BTN3A1 KO 293T cells (293T cells with CRISPR-Cas9 inactivated *BTN3A1* gene only) and tested for PAg-dependent activation of human and alpaca TCR transductants, drawing a complex picture of synergism and the interference of the different BTN3 molecules. At first, we confirmed that the expression of BTN3A1 in BTN3KO cells rescued PAg reactivity only poorly, while the response of human TCRs to BTN3A1 KO cells transduced with BTN3A1 was fully reconstituted. Despite a massive overexpression of BTN3A1, compared to endogenous expression in wild-type 293T cells, PAg stimulation and the degree of Vγ9Vδ2 T cell activation was the same. The transduction of BTN3KO cells with alpaca BTN3 induced a PAg response of vpTCR (alpaca TCR) but not of huTCR (human TCR–MOP) transductants, indicating some species specificity of the TCR–BTN3 interaction. Surprisingly, BTN3A1 KO cells transduced with vpBTN3 stimulated neither human nor alpaca TCR-transduced cells. This may indicate an interference of human BTN3A2 and BTN3A3 molecules with molecules mandatory for effective PAg sensing, e.g., human BTN2A1, whose function will be discussed in the next paragraph. BTN3KO as well as BTN3A1 KO cells transduced with a human BTN3A1–ED/vpBTN3–TM/ID chimera was very similar and stimulated human and alpaca TCR transductants even better than wild-type 293T cells, suggesting that the vpBTN3–TM/ID substitutes the “help” by BTN3A2/3 and that the presence of endogenous BTN3A2/BTN3A3 had no effect. Quite dramatic differences were seen for chimeras of alpaca BTN3–ED and human BTN3A1–TM/ID. If transduced in BTN3KO cells, they stimulated neither human nor alpaca TCR-expressing reporter cells, but after expression in BTN3A1KO cells, they activated human as well as alpaca TCR transductants. Our interpretation is that PAg binding to the human ID of this chimera does not lead to the changes required to induce the “activating” conformation of the alpaca BTN3–ED, but this capacity somehow translates to BTN3A2 and/or BTN3A3, which can then be sensed by either the human or the alpaca TCR [20]. 

As shown in Figure 3, the amino acid sequence comparison of vpBTN3 with human BTN3A1/A2 or A3 revealed a very similar degree of identical amino acids for the three molecules (without leader sequence). This similarity included the V domain, C domain, and the B30.2 of BTN3A1 and BTN3A3, respectively. Only for vpBTN3–TM the similarity to BTN3A1 (88%) was clearly higher than for BTN3A2 and BTN3A3 (both 76%), while the respective percentage of identity for the JM domain was 64% for BTN3A3 and 49% for BTN3A1. We hypothesize that the BTN3A3-like nature of the alpaca JM domain enables efficient PAg sensing by a single molecule. 

## 5. BTN2A1: A Missing Link Poses Questions

### 5.1. BTN2A1, a New Player in the Game: Identification of BTN2A1 as a Prerequisite of PAg Recognition

BTN3A1 is a key compound in PAg sensing and γδ T cell stimulation. Nevertheless, Harly et al. already discussed that ABP-pulsed BTN3A1-transduced mouse cells failed to stimulate Vγ9Vδ2 T cells [53]. However, the interpretation of such negative results is difficult, since they might reflect reduced co-stimulatory signals or cell–cell adhesion as a consequence of the species barrier between stimulating and responding cells. The use of the murine reporter system described above allowed an approximation by demonstrating that APC–reporter cell interaction was at least sufficient to induce another type of TCR-mediated response, namely the induction of a peptide-specific response via an αβ TCR [53]. More precise was the demonstration of a mAb 20.1-induced response by TCR–MOP transductants to rodent APCs (mouse or hamster cells) transduced with BTN3A1 and CD80, while no activation by PAg could be observed [95]. Importantly, CD80-transduced hamster cells containing human chromosome 6 (CHO-Chr:6) stimulated reporter cells in the presence of HMBPP or Zol. Furthermore, after Zol pulsing, the same CHO-Chr:6 cells induced CD69 upregulation of human peripheral blood Vγ9Vδ2 T cells, which was increased by BTN3A1 expression, while BTN3A1-transduced CHO cells without Chr:6 had no effect. This allowed the interpretation that not only *BTN3A1* but also other genes on Chr:6 were necessary for PAg-induced Vγ9Vδ2 T cell activation [95]. Obvious candidates were *BTN3A2* and *BTN3A3*, but their transfer to CHO cells was not sufficient to induce a PAg-specific response, albeit higher “background” stimulation was found (Paletta, Fichtner, Karunakaran, Herrmann, unpublished data). 

An interesting aspect is that PAg stimulation required only minimal expression of BTN3A which could be below the detection limit of mAb 103.2 in flow cytometry while stimulation by mAb 20.1 correlated with the intensity of BTN3A expression [96].

In order to identify the additional gene(s) controlling the species-specific PAg-mediated activation, we generated radiation hybrids (RH) and tested them for the phenotype “PAg-dependent stimulation” as we did for the analysis of the CHO-Chr:6 monosomal line [84,95]. Radiation hybrids are somatic cell hybrids of irradiated (human) cells and a hypoxanthine-aminopterin-thymidine (HAT)-sensitive (rodent) cell line [97,98]. The chromosomes of the donor genome are fragmented by the irradiation and parts of the fragmented genome become integrated into the recipient genome in the hybrid cell (e.g., parts of human Chr:6 in CHO–Chr:6 cells). The proportion of integrated genome ranges from 10% to 50% and the chromosomal fragment size decreases with the increase of the irradiation dose. We generated RHs with BTN3A1 + CD80-transduced HAT-sensitive hamster or mouse cells as fusion partners for irradiated CHO–Chr:6 cells. This allowed testing the BTN3A1-dependent stimulatory capacity of the RHs with mAb 20.1. The transduction of BTN3A1 also prevented the “rediscovery” of BTN3A1 as a mandatory gene for PAg stimulation and increased the chance of obtaining stimulatory RH. As described in the study [68], we characterized six stimulatory RHs in greater detail. Their genomic content was deduced from comparing their transcriptomes to those of both fusion partners with CHO–Chr:6 as positive control and the rodent fusion partner as negative control. This comparison identified a candidate region of 580 kB on Chr:6, which contained a high number of histone genes, tRNA, and other non-membrane protein genes. The only membrane protein-encoding genes were those of the *BTN* cluster and the highly conserved MHC class I-like iron transporter HFE. *HFE* and all *BTN* genes with the exception of *BTN1A* were expressed by all stimulatory RHs. Since BTN3As alone do not reconstitute the PAg response, BTN2A1 and BTN2A2 were prime candidates for the missing Chr:6 encoded gene(s), contributing the PAg presentation phenotype. The functional deletion of BTN2A1 and BTN2A2 in 293T cells as well as of BTN2A1 alone abolished the capacity of Zol-pulsed cells to stimulate human γδ T cells and reporter cells in a Zol-, HMBPP-, or mAb 20.1-dependent manner. The knock-out of BTN2A2 had no such effect [68]. No negative effects of the loss of BTN2A2 for ABP-induced stimulation were observed in BTN3A1 + BTN3A2-reconstituted *BTN3A* knockout cells, which lack the *BTN2A2* gene as a consequence of the knock-out strategy [72]. 

Independently, Rigau et al. identified BTN2A1 as a mandatory component of PAg presentation by screening cells transfected with an shRNA library for the reduction of Vγ9Vδ2 TCR tetramer binding and identified signal peptide peptidase-like 3 (SPPL3) and BTN2A1 as prime candidates. Knock-out of BTN2A1 in different human cells massively reduced the binding of TCR tetramers and newly generated BTN2-specific mAbs inhibited tetramer binding as well as stimulation with ABP [19]. 

### 5.2. BTN2A1: Interaction with TCR and BTN3A1 

Both groups found that rodent cells transduced with BTN2A1 and BTN3A1 reconstituted the PAg response and TCR tetramer binding. Importantly, TCR tetramer binding relied only on BTN2A1 expression, while BTN3A1 showed no direct TCR interaction. The treatment of BTN3A1 + BTN2A1 transductants with ABP did not increase TCR tetramer binding, suggesting that BTN2A1 binds the TCR autonomously, while both BTNs were required for the induction of a PAg-dependent Vγ9Vδ2 T cell response [19,68]. The direct binding of BTN2A1 to the Vγ9 TCR was demonstrated by plasmon resonance studies. Similar K_D_s of about 50 µM were demonstrated and were independent of the CDR3s of Vγ9 and the paired δ-chain. Molecular modeling and mutagenesis studies mapped the surface formed by the C-F-G strands of BTN2A1 (C-F-G surface) as the contact region of the TCR. Interestingly, this area was previously mapped by Willcox et al. for the binding of BTNL3 to the human Vγ4 chain [18]. An important contribution of the germline-encoded HV4 of Vγ9 was found by testing the binding in plasmon resonance studies and in functional assays for PAg responses [19,68]. Yet, the interaction of BTN2A2 and the TCR remains unclear. In plasmon resonance studies, the affinity of recombinant BTN2A2 to the Vγ9Vδ2 TCR was even higher than that of BTN2A1, while TCR tetramers showed nearly no binding to BTN2A2 transductants. In part, this difference may be explained by the different degree of cell surface expression in these experiments [68]. Another observation of unclear significance was a disulfide bridge that stabilized BTN2A1 as a homodimer formed by a cysteine at position 247, which is unique to BTN2A1. All other BTNs and BTNL molecules carry a tryptophan at this position, and a C247W substitution of BTN2A1 had no negative effect on PAg stimulation or TCR tetramer binding.

The interaction of the Vγ9-HV4 with the C-F-G surface was confirmed by mutational analysis and tested for effects on PAg or ABP stimulation. A mutation of a negatively charged glutamic acid to an alanine in the Vγ9-HV4 (E70A) reduced the PAg response of TCR–MOP transductants [68] and the Zol-induced CD69 expression of TCR-transduced Jurkat cells [19]. Furthermore, mutations of the positively charged amino acids arginine or lysine largely (E70R) or completely (E70K) abolished the PAg-induced stimulation [68]. Interestingly, the extent of reduction in TCR binding as measured by binding of BTN2A1 tetramers to TCR mutant cells was much more dramatic than the reduction of the TCR-mediated IL-2 production in response to HMBPP or Zol-induced CD69 expression. Importantly, mutating the Vγ9 CDR3 and Vδ2 CDR2 and CDR3 had no effect on BTN2A1 binding, but stimulation by Zol or HMBPP was lost [19,68]. 

BTN2A1 interaction with BTN3A1 was also demonstrated by confocal microscopy and FRET [19], by immunoprecipitation after cross-linking cells with a water-soluble membrane-impermeable chemical crosslinker, and by NMR studies [68]. Similar to the BTN2A1–TCR interaction, interaction between the V domains of both molecules at the surface formed by their C-F-G surfaces could be predicted from NMR studies [68]. Such C-F-G-interface interaction between V domains of members of the Ig-superfamily and explicitly of the B7 family is not uncommon, as shown for cis-interaction of PD-L1 (CD274) and B7.1 (CD80) [99]. Importantly, mutations of BTN3A1 and BTN3A2 in this area reduce PAg stimulation, which is a result that was considered evidence for a potential TCR-binding site but may now warrant new interpretation, since BTN3A–BTN2A1 might also be affected. The same may apply for the inhibition of PAg responses by mAb 20.1, which binds to the surface formed by C-C´-C´´ strands and may also affect the neighboring the C-F-G surface (Figure 3) [18,68]. It is conceivable that mAb 20.1 binding interferes with BTN2A1–BTN3A1 interactions and consequently Vγ9Vδ2 T cell responses but also that a conformational change as a consequence of mAb 20.1 binding might propagate TCR binding and the activation of certain Vγ9Vδ2 TCR clonotypes while interfering with a stimulatory state after PAg binding to BTN3-ID. In line with this idea, single-chain variable fragment (sc-Fv) of the non-competing mAb 103.2 show no inhibitory effects [54], despite a high affinity for recombinant BTN3A1, 2 and 3 (K_D_ 8-15 nM). Nonetheless, the whole 103.2 antibody is a very efficient inhibitor [57] perhaps by fixing a conformation of the BTN3A molecules, which prevents interaction with TCR, BTN2A1, or other not yet defined molecules. 

### 5.3. A Composite Ligand Model of PAg Recognition

When discussing binding partners of the TCR, it is of interest that BTN2A1- as well as BTN2A1 + BTN3A1-transduced murine or hamster cells showed a robust and statistically significant background stimulation of 1–10% of the maximum response to HMBPP. This was completely abolished by mutations of the Vγ9-HV4 but also by mutations/deletions in the CDR2δ and CDR3δ [68]. Although these regions of the TCR δ-chain were not involved in BTN2A1 binding [19,20], both are well established to contribute to the HMBPP or Zol response [100]. We suggest that these TCR regions may interact or bind to other proteins beyond BTN2A1, which are required for full activation of the Vγ9Vδ2 T cell (Figure 6) and hypothesize a composite ligand model of PAg recognition. 

Such additional binding would be sterically possible and may also contribute to PAg sensing or in certain cases activate Vγ9Vδ2 T cells in a PAg-independent manner. Some of these ligands may have an affinity to the Vγ9Vδ2 TCR that is too low to trigger a response on their own and may need to associate with BTN2 or BTN2–BTN3 complexes to be sensed by the Vγ9Vδ2 TCR. Others may have a higher intrinsic affinity for specific TCR clonotypes and may be able to interact with the TCR even in the absence of PAg-induced cellular changes [68] but in some cases may still require “help” by the superantigen-like binding of BTN2A1 [21]. 

In this case, the CDR3s of the TCR γ- and δ-chain would be of importance to generate a potent PAg response but also for the reaction to protein antigens. Such additional ligands could explain the differential PAg responses to certain tumor cells as reported by the Kuball group [101]. It could also explain the reactivity of Vγ9Vδ2 TCR-G115 to mitochondrial F1-ATPase, which is ectopically expressed on the cell surface of certain tumors, for which binding to the TCR-G155 has been reported [36]. In this case, TCR-G115 could have a weak intrinsic reactivity to F1-ATPase, which might be enhanced by endogenous PAg and BTN2A1 and/or BTN3A1. Under these circumstances, BTN3A1 would not necessarily need to be a TCR-binding molecule but could act as chaperone positioning the additional protein ligand [102]. 

### 5.4. BTN3A1-Expressing Rodent Cells as Mediators of mAb 20.1-Induced Vγ9Vδ2 T Cell Activation 

An open question is why mAb 20.1 induces an activation of Vγ9Vδ2 TCR transductants by BTN3A1-transduced rodent cells, which apparently lack BTN2A1 [54,68]. The easiest explanation would be that rodent BTN2A2 can to some extent replace human BTN2A1 in mAb 20.1-dependent stimulations. This is not unlikely, since both molecules show a high degree of similarity of their ED-V domain but differ in their TM and ID, which may explain why BTN2A2 cannot support the PAg-sensing mechanism of BTN3A1 in the same manner as human BTN2A1. These and other open questions on the functionality of BTNs could be addressed by the analysis of BTN2A2-deficient rodent cells, BTN2 mutants, and chimeras. The importance of the BTN2A1–ID for PAg sensing has already been highlighted by the work of Rigau et al. who demonstrated that deletion of the BTN2A1–ID leads to a loss of PAg- and mAb 20.1-induced activation but not of TCR tetramer-binding [19]. 

The transduction of rodent cells (mouse and hamster) with BTN2A1 and BTN3A1 allows efficient PAg-dependent activation. The additional transduction of BTN3A2 had no effect in the case of mouse 3T3 cells and only modest effects in the case of transduced hamster cells, especially if compared to the dramatic difference seen between BTN3A1- or BTN3A1 + BTN3A2-reconstituted BTN3KO 293T cells [19]. One possible explanation for the lack of demand of BTN3A cooperation in rodents could be that in human cells, BTN3A1 as a mediator of PAg sensing might be inhibited by an additional factor that provides strict control on Vγ9Vδ2 T cell stimulation and which might be functionally neutralized by BTN3A2 or BTN3A3. In rodents that lack the PAg-sensing triad (*BTN3, TRGV9* and *TRDV2*), such tight control may be superfluous. 

### 5.5. BTN3A–TCR Interaction: Yes, No or Yes and No?

BTN3A1 is without any doubt a key player in PAg sensing and Vγ9Vδ2 T cell activation, yet, its interaction with the TCR is still an open question. Attempts of several groups to measure the binding of Vγ9Vδ2 TCR to BTN3A1 by ITC [67], NMR [67], or TCR tetramers [68] were without success, although other weak interactions such as that of BTN3A1–ID with PAg [67] or the TCR with BTN2A1 [68] or of BTN2A1 with BTN3A1 [68] were detected. The only study reporting evidence of binding of recombinant BTN3A1–ED to the TCR and HMBPP is heavily refuted [66]. Partial evidence for a BTN3A1–TCR interaction is a crystal structure of the BTN3A1 V domain with HMBPP and IPP generated by the De Libero group [66]. These data have been criticized as a misinterpretation of electron dense material in the “binding groove”, which in fact may by polyethylene glycol used for crystal generation [67]. There are also other experiments suggesting the binding of PAg to recombinant BTN3A1 V domains, but whether this binding is specific and directly related to PAg-mediated stimulation is difficult to judge without crystallographic data and reproduction of some functional data on the activation of Vγ9Vδ2 TCR-transgenic cells by immobilized BTN3A1. The same paper describes BTN3A1–TCR interactions by plasmon resonance analysis for Vγ9Vδ2 TCR dextramers and immobilized BTN3A1–ED as well as by surface-enhanced Raman scattering. Both approaches suggest a very low affinity for Vγ9Vδ2 TCRs, while no binding was found for other γδ or αβ TCRs. The presence of IPP increased this interaction by reducing the K_D_ three-fold [66]. These data of the Rossjohn laboratory may indicate a very low “intrinsic” affinity of BTN3A1 to the TCR, which could still contribute to Vγ9Vδ2 T cell activation [66]. Therefore, it would be of considerable interest to test whether the mutations of BTN3A1/2 ED, which are reported to affect BTN3A1/2-mediated PAg stimulation, also affect this weak BTN3–TCR interaction. Nevertheless, all published experiments of other groups speak against BTN3A molecules as antigen-presenting molecules. 

Despite everything, even if, as we hypothesize, BTN3 might serve as a chaperone that brings another ligand (e.g., evolutionary highly conserved proteins) to the cell surface or positions it to the TCR, this would not exclude a direct binding of BTN3A to the Vγ9Vδ2 TCR in a certain conformation or in complex with other molecules. 

## 6. Future Directions

Despite the considerable efforts made to solve the puzzle of PAg-mediated Vγ9Vδ2 T cell activation, which led to the identification of compounds involved in this process and the detailed analysis of the structure–activity relationship of phosphoantigens and BTN3s, several basic questions are not yet answered. This is true for the BTN3–TCR interaction, the identification of the “active” form of BTN3A1, and the mechanism underlying the partial mimicking of PAg by agonistic BTN3-specific antibodies. 

For some of those open questions, experimental approaches using biochemical and biophysical methods are likely to provide clear answers such as the structural basis of interaction between BTN2A and TCR or between the extracellular domains of BTN3 and BTN2. Other problems such as the identification of the hypothesized CDR3 and TCR δ-chain ligand will be more difficult to answer. However, species comparison studies can provide new perspectives as demonstrated by the analysis of alpaca BTN3 as a PAg sensor or the identification of BTN2A1 as new players through methods exploiting species differences as in the case of radiation hybrids. As discussed in the review, species differences can be used for further molecular analysis e.g., by analyzing chimeric molecules or as a guide for mutational analysis and for identifying features of BTN2 molecules that might be conserved in species with functional *TRGV9* genes. 

Mice lack *TRGV9* genes [85] and have no functional BTN2A1 ortholog but a functional *Btn2a2* gene [103]. For the *Btn2a2-/-* mouse enhanced CD4 and CD8 T cell responses were reported and attributed to APCs rather than T cells or non-hematopoietic cells [104]. γδ T cells were not specifically addressed in this study but mouse and human BTNs have been shown to modulate the immune response for *in vitro* and *in vivo* models [58,103,105]. In conclusion, BTN family members control and support γδ T cell development and activation on the one hand and on the other hand modulate immune responses in a more general manner. It will be interesting and feasible to dissect these dual functions on a molecular level, and species comparisons might be helpful for this purpose.

Another important task is establishing mouse models for Vγ9Vδ2 T cells. A Vγ9Vδ2 TCR transgenic mouse has been described by the De Libero group [66], but this model organism seems to have a block in thymic development, which can be overcome by the application of anti-CD3 antibodies that may mimic positively selecting ligands [106]. Therefore, it is tempting to speculate that BTN3As and/or BTN2A1 either on their own or together with highly conserved proteins might serve as such ligands and also maintain the post-thymic expansion of Vγ9Vδ2 T cells. This would be similar to the role of BTN heterodimers such as BTNL3/8 and BTNL1/6 for subsets of human or mouse γδ T cells for the maintenance and expansion of intestinal T cells [15]. The generation of mice transgenic for pairs of human BTNs and the respective BTN-binding TCRs might pave the way to study the development and physiological function not only of PAg-reactive but also other human γδ T cells in a small animal model and to test their potential in combatting tumors and infectious diseases. In addition, those mice could be used as a preclinical model for γδ T cells activating or modulating agents such as BTN-specific monoclonal antibodies. 

## Figures and Tables

**Figure 1 cells-09-01433-f001:**
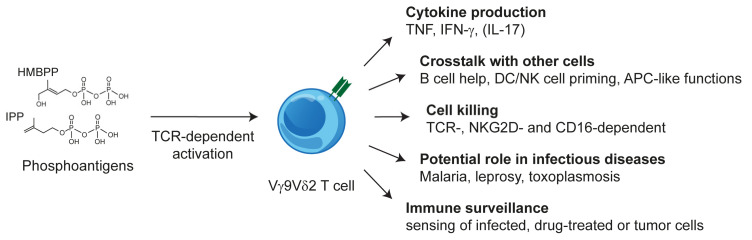
The multifunctionality of Vγ9Vδ2 T cells.

**Figure 2 cells-09-01433-f002:**
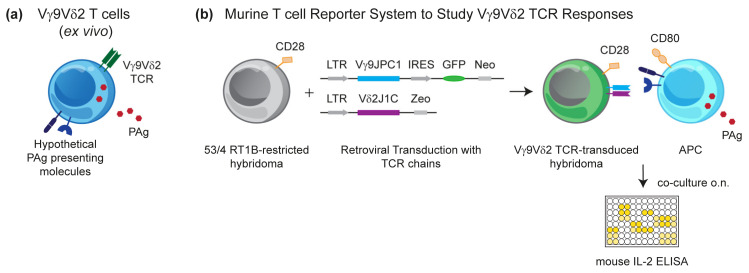
Generation of a murine reporter T cell system. (**a**) Hypothetical phosphoantigen (PAg)-presenting molecules are believed to be expressed ubiquitously and human Vγ9Vδ2 T cells could potentially also present PAg. (**b**) This drawback could be overcome by using murine reporter T cells, which lack an endogenous PAg response. Murine T cell antigen receptors (TCR)-negative 53/4 RT1B-restricted hybridoma T cells were transduced with human *TRGV9JPC1* (Vγ9JPC1) and *TRDV2J1C* (Vδ2Jδ1C) constructs encoding Vγ9 and Vδ2 TCR chains, respectively. For co-stimulation, CD28 was overexpressed in hybridoma T cells, and endogenous CD3 enabled TCR complex formation. Thus, generated 53/4 Vγ9Vδ2 TCR hybridoma cells could be activated in the presence of PAg when co-cultured with CD80-transduced antigen-presenting cells (APCs) of human origin or other species, provided they are expressing the molecules necessary for PAg presentation. Mouse interleukin (IL)-2 produced by the T cell hybridoma in overnight co-cultures was measured as read-out for reporter T cell activation.

**Figure 3 cells-09-01433-f003:**
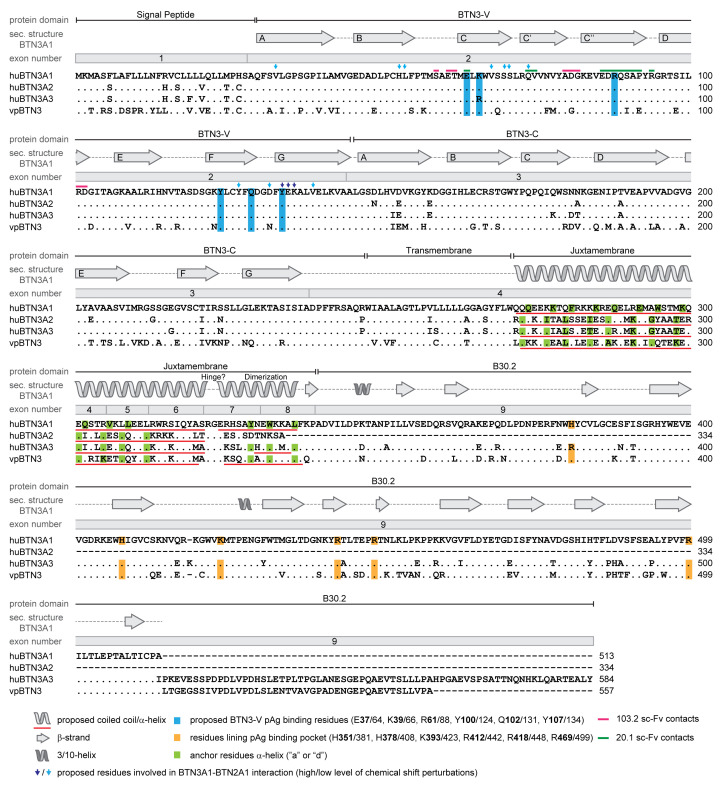
Sequence comparison of human and alpaca butyrophilin (BTN) 3 isoforms. Protein sequence alignment of human BTN3A isoforms and alpaca BTN3 (vpBTN3) (Clustal Omega and BioEdit). Identical amino acids (dots), gaps (dashes), and numbers (right side) are indicated. Amino acid numbers include the signal peptide and are written in standard letters. For amino acid numbers used in the main text, the leader peptide was left out. They are given in bold. The domains of BTN3 (protein domain) and corresponding exons (exon number) are shown according to human BTN3A1 [57,64,65]. Some features of the secondary structure of BTN3A1 are visualized (according to [57,65] and the Protein Data Base BTN3A1 B30.2 protein structure 5HM7). Predicted coiled-coil structures in the juxtamembrane domain (JM) of all BTN3s are shown (red) together with predicted anchor residues (light green) as proposed by Wang et al. [65]. Proposed PAg-binding residues in the BTN3A1 V domain (BTN3-V) are highlighted in blue [66] and residues lining the PAg binding pocket of B30.2 are shown in orange [67]. Contact regions of sc-Fv 103.2 (pink line) and 20.1 (dark green line) in the BTN3A1 V domain are indicated [57]. Blue arrows indicate the residues undergoing chemical shift perturbations (CSP) in a nuclear magnetic resonance (NMR) study of the BTN3A1–BTN2A1 interaction [68]. The residues with the most pronounced CSPs are marked with dark blue arrows, whereas lower CSPs are indicated by light blue arrows.

**Figure 4 cells-09-01433-f004:**
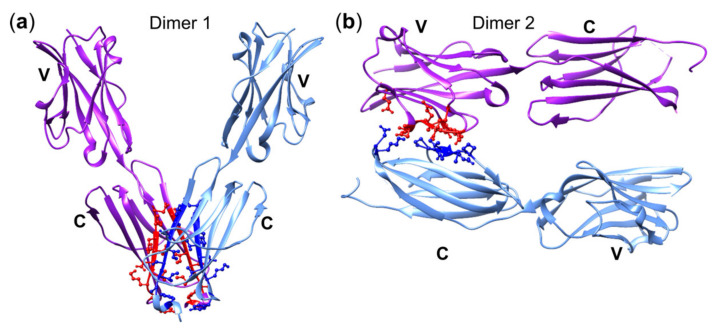
BTN3A ectodomain dimers. (**a**) Dimer 1: The BTN3A–EDs interact at their C domains to form a V-shaped dimer. Interacting residues are highlighted in red and blue. (**b**) Dimer 2: The V domain of a BTN3A–ED interacts with the C domain of another BTN3A–ED, forming a head-to-tail conformation. Interacting residues are highlighted in red (V domain) and blue (C domain). Contact residues of either domain [57] are according to the PDB protein structures 4F9L and 4F80 and were modified with USCF Chimera1.1.

**Figure 5 cells-09-01433-f005:**
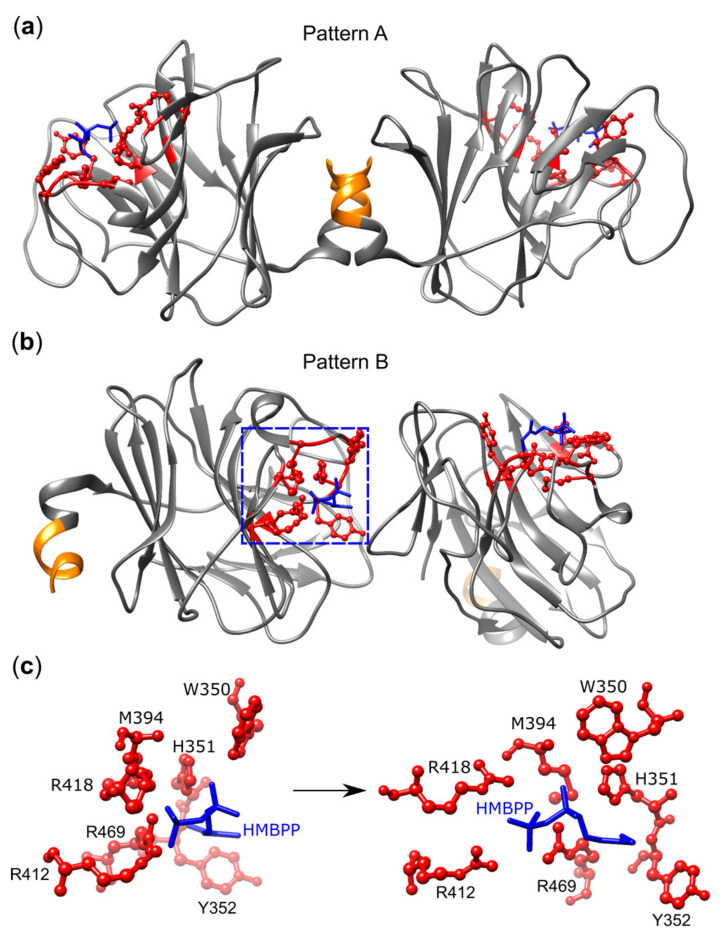
(*E*)-4-hydroxy-3-methyl-but-2-enyl pyrophosphate (HMBPP) binding to the BTN3A1 B30.2 domain. (**a**) Pattern A: Symmetric dimer formation by two B30.2 domains of BTN3A1 in HMBPP-bound form. HMBPP-B30.2 interfaces were positioned as far from each other as possible. (**b**) Pattern B: HMBPP-bound B30.2 domain forms asymmetric dimers where HMBPP and its interacting residues of a monomer are positioned at the interface of another monomer whose HMBPP interface is oriented to the opposite side. (**c**) Islet of pattern B representing HMBPP-interacting residues of the B30.2 monomer (left) and a changed orientation to obtain the most planar view (right). The JM is shown as an orange helix, positively charged residues surrounding the HMBPP pocket are shown in red, and HMBPP is shown in blue. For this figure, the PDB protein structure 5ZXK [74] was modified with UCSF Chimera 1.1.

**Figure 6 cells-09-01433-f006:**
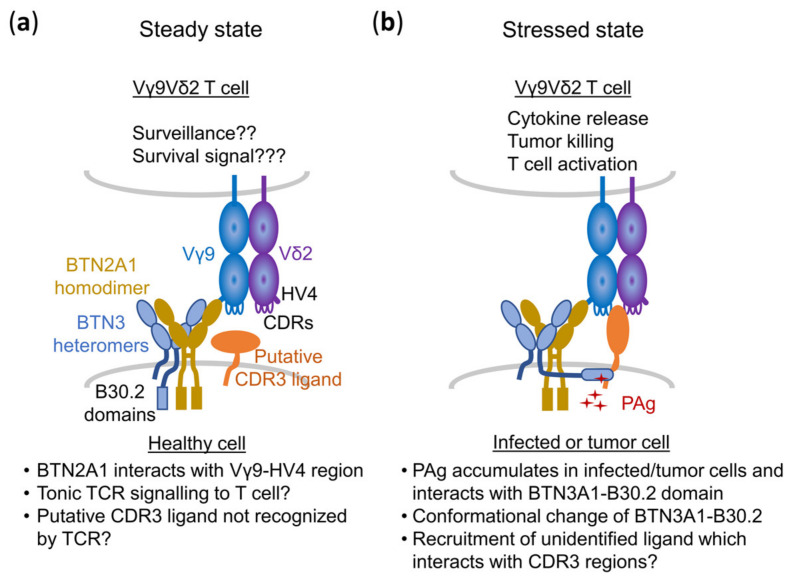
Composite ligand model of PAg sensing by the Vγ9Vδ2 TCR. (**a**) Steady state: A healthy APC would communicate with Vγ9Vδ2 T cells through an autonomous interaction between the BTN2A1 V domain and the Vγ9–HV4 region. This interaction might provide tonic signals necessary for the surveillance and survival of T cells without the involvement of the Vδ2 chain. Simultaneously, BTN3A tends to remain associated with BTN2A members on the APC via their V domains with unlikely interaction of BTN3A1 with the TCR (**b**) Action of PAg: In the presence of stress or PAgs, a PAg-dependent activation complex would be assembled on the APC. This complex could be inclusive of mandate BTN2A1 and BTN3A1 members, and other unidentified putative membrane proteins or ligands presumably recruited by BTN2/BTN3 members as a consequence of PAg interaction with the B30.2 domain of BTN3A1. Such an activation complex interacts with Vγ9Vδ2 TCR, which is certainly dependent on all CDRs of Vγ9 and Vδ2 chains (right). Picture modified after [68].

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
