# Peer review of "An Update on the Molecular Basis of Phosphoantigen Recognition by Vγ9Vδ2 T Cells"

_cells, 2020, doi:10.3390/cells9061433_

Round 1
Reviewer 1 Report
First of all, a hearty thanks to the authors for a highly enjoyable afternoon reading their fine paper. There could not have been a more qualified team to tackle this subject! The paper clearly reflects this expertise and long-standing involvement.
The article aptly describes the current state of knowledge about the human gd T cell response to pAgs, which was discovered roughly two and a half decades ago and continues to command a great amount of interest, not only because it represents a fascinating problem for immunologists but also because of its likely relevance in tumor immunology and non-conventional vaccine design.
Over the years, an enormous effort has been dedicated to gain an understanding of this response at cellular and molecular levels, and much progress has been made. And yet, as clearly outlined in this manuscript, some of the most basic questions remain unanswered: Is there a direct cognitive interaction between pAgs and TCRs - which would justify calling these phosphate-moieties "antigens"? How many distinct molecular components are required to enable recognition of pAgs, and how might these molecules interact and trigger intra-cellular signalling? Is pAg-recognition the main "purpose" of Vg9Vd2 cells or rather an auxiliary sensitivity capable of enhancing more clonal antigen responses? etc.etc.
No doubt, this article is very well suited as an advanced level introduction to the topic and should be read by students in the field. However, I expect that many others will be attracted also, and this leads me to mention a few minor concerns.
- I found the title misleading as it suggests that the problem is solved when in fact it is not.
- The two introductory paragraphs are lengthy and not directly connected with the focus of the paper - I would have preferred to see a more detailed introduction of the pAg response - discovery, in vivo relevance, potential uses etc.
- Some of the Figs. could be improved: Fig.1 - "expansion..." does not fit well with the functional responses listed. Fig.2 - parts a) and b) should be more clearly separated. Fig.6 - should be divided into parts a) and b) with subtitles. The concept of tonic signalling is highly interesting and might connect gd T cells with the other lymphocyte types, which also engage in tonic signalling. But it seems unclear if only signalling without pAgs fits into this category or if signalling with pAgs might be just an enhanced form of tonic signalling.
- There seems to be a disconnect between the 9 secs it takes for BrHPP-exposed gd cells to respond (line 152) and the 5 min it takes before binding between BTN3A1 and TCR becomes detectable (lines 306-310). Is this merely an assay problem?
- Some minor language issues: line 36("compounds" should be components), line 448 (TCRs do not "respond", cells do), line 711 (BTN2A1-C-F-G appears here but -C-F-G is only defined later, partially in line724)
- Too much? It seems that the final paragraphs, starting with the introduction of BTN2A1 (line 636) are not yet ready for prime time - at least, they should be shortened and their speculative nature pointed out more clearly, which brings me back to point 1: The problem is not yet solved and the title of the article should reflect this circumstance.
Author Response
First, we would like to thank the reviewer for his kind comments. We appreciate it.
We uploaded a revised manuscript (wirth new line numbers). Here or response to the comments.
- I found the title misleading as it suggests that the problem is solved when in fact it is not. Title has been changed. Following the suggestion of one of the other referees we removed also “human” from the title and rephrased it.
- The two introductory paragraphs are lengthy and not directly connected with the focus of the paper - I would have preferred to see a more detailed introduction of the pAg response - discovery, in vivo relevance, potential uses etc.
Has been shortened and a short introduction referring to excellent reviews in this and other journal covering the topics has been inserted.
- Some of the Figs. could be improved: Fig.1 - "expansion..." does not fit well with the functional responses listed. Has been changed
- Fig.2 - parts a) and b) should be more clearly separated. Has been changed
- Fig.6 - should be divided into parts a) and b) with subtitles. The concept of tonic signalling is highly interesting and might connect gd T cells with the other lymphocyte types, which also engage in tonic signalling. But it seems unclear if only signalling without pAgs fits into this category or if signalling with pAgs might be just an enhanced form of tonic signalling. We modified the figure accordingly. With respect to PAg, we would prefer to leave it as it is in order to keep the figure simple. It´s a worling hypothesis (and one of many possible ones) anyway.
- There seems to be a disconnect between the 9 secs it takes for BrHPP-exposed gd cells to respond (line 152) and the 5 min it takes before binding between BTN3A1 and TCR becomes detectable (lines 306-310). Is this merely an assay problem? We don´t know. It could be an assay problem or being more significant. Therefore we just refer to the published data
- Some minor language issues: line 36("compounds" should be components), line 448 (TCRs do not "respond", cells do), line 711 (BTN2A1-C-F-G appears here but -C-F-G is only defined later, partially in line724) Corrected
- Too much? It seems that the final paragraphs, starting with the introduction of BTN2A1 (line 636) are not yet ready for prime time - at least, they should be shortened and their speculative nature pointed out more clearly, which brings me back to point 1: The problem is not yet solved and the title of the article should reflect this circumstance. We made small changes in this part and extended the part on future directions. The new title should make clear that understanding PAg-action is “work in progress”. .
Reviewer 2 Report
Excellent review of this yet puzzling mechanisms. State of the art positioning of current knowledge, models, and reagents. Very nice figures, almost exhautive bibliography cited.
Very few and minor edition revisions:
-line 118: I suggest to quote the original paper introducing/defining "phosphoantigen": Fournie, J.J., and Bonneville, M. (1996). Stimulation of gamma delta T cells by phosphoantigens. Res Immunol 147, 338-347.
-line 135: original refs for nucleotide PAg are far older: Poquet, Y., Constant, P., Halary, F., Peyrat, M.A., Gilleron, M., Davodeau, F., Bonneville, M., and Fournie, J.J. (1996). A novel nucleotide-containing antigen for human blood gamma delta T lymphocytes. Eur J Immunol 26, 2344-2349.
or
Constant, P., Davodeau, F., Peyrat, M.A., Poquet, Y., Puzo, G., Bonneville, M., and Fournie, J.J. (1994). Stimulation of Human Gamma-Delta T-Cells by Nonpeptidic Mycobacterial Ligands. Science 264, 267-270.
-line 183-184: the sentence seems to lack a verb
-line 253: delete one "induced"
-line 377: replace "adapted" by "adopted"
-line 808: correct "may by " with "might be"
Author Response
Thank You for your kind comments. Corrections have been made and thank you for pointing out the missing references.
Reviewer 3 Report
The review article from Herrmann and colleagues describes the molecular basis of phosphoantigen recognition by gamma delta T cells. This manuscript is from a true leader in the field. As such, the manuscript is quite well written, the references are outstanding, and the level of detail provided is also outstanding. The manuscript focuses on the key proteins involved in phosphoantigen detection, specifically the gd TCR and BTN3/2. Through studies with different antibodies, molecular constructs, and different species, the authors have put together a comprehensive and detailed characterization of the molecular interactions involved in this process.
In particular, the Figure 3 of this paper is a great figure that should be quite useful to the field, because it compiles much of the discussion into one key figure. However, there are minor inaccuracies and some improvements that could be made here. Most importantly, the authors should note that the AA numbering in Figure 3 does not match the rest of the paper. This is because BTN3A1 has a 30 AA signal peptide that is removed in the mature protein. A problem developing in the field is that the structural biologists have multiple publications numbering the protein without the 30 AA signal peptide, while the cell biologists have numbered the protein starting with the start codon. Figure 3 numbers from the start codon, but in the legend some residues are listed from the mature protein. For example, H351 should be H381 in the MSA, etc. This needs to be resolved by using one numbering scheme or the other, or perhaps listing both of them.
Also in Figure 3: 1) the signal peptide could be labeled. 2) Does the B30.2 domain really extend to the very end of the proteins? 3) is the TM/JM really a coiled coil structure- there seems to be some dispute about this and the sequence is not that of a classical coiled coil. 4) is there any reason to include BTN2 in this alignment?
From a broader perspective, the paper does a great job in discussing the evolutionary relationships of these proteins, and this is a big section of the review article. Its also a unique angle. However, I thought this angle could be more emphasized in the abstract, as well as the Future Directions, which surprisingly did not mention any future directions related to the evolution of the complex.
Likewise, while there is much discussion on the triad of BTN3, TCRG, and TCRD with respect to other species, there was no discussion of BTN2 in this context. Perhaps this has not been studied yet, but I was hoping to know more about BTN2 in these species and left hanging.
The future directions could mention what is still left unresolved about the mechanism, as the mechanism was a key part of the paper.
Should "human" be in the title? The paper does such a great job talking about other species, its really more broad than just human.
Minor points:
- Lines 12 and 120 - Hydroxy need not be capitalized, common names of chemicals can be lower case unless starting a sentence.
- Line 130 - FPPS is used before its defined, and that whole sentence is a bit awkward.
- Line 203 - Should be "B30.2 (or PRY/SPRY)"
- Line 355 - technically not all small GTPases are "activated" by NBPs. NBPs inhibit PT processing of all prenylated GTPases, sometimes the result is increased GTP binding and sometimes it decreases. Even if increased, does not necessarily mean its "active" as it may be mislocalized.
- Line 582 - Crisp should be CRISPR.
Author Response
We would like to thank for the kind and constructive comments. Here our response:
The future directions could mention what is still left unresolved about the mechanism, as the mechanism was a key part of the paper.
We extended this part and hope that it will be helpful. Writing more about BTN2s and species would be rather speculative and beyond the format of this review.
Should "human" be in the title? The paper does such a great job talking about other species, its really more broad than just human.
You are right. We removed human
Minor points:
- Lines 12 and 120 - Hydroxy need not be capitalized, common names of chemicals can be lower case unless starting a sentence. Corrected
- Line 130 - FPPS is used before its defined, and that whole sentence is a bit awkward. Has been rewritten lines 193-195
- Line 203 - Should be "B30.2 (or PRY/SPRY)". Corrected
- Line 355 - technically not all small GTPases are "activated" by NBPs. NBPs inhibit PT processing of all prenylated GTPases, sometimes the result is increased GTP binding and sometimes it decreases. Even if increased, does not necessarily mean its "active" as it may be mislocalized. Has been rewritten lines 465-467
- Line 582 - Crisp should be CRISPR. Corrected